# Screening and Verification of Reference Genes for Analysis of Gene Expression in Garlic (*Allium sativum* L.) under Cold and Drought Stress

**DOI:** 10.3390/plants12040763

**Published:** 2023-02-08

**Authors:** Qizhang Wang, Chunqian Guo, Shipeng Yang, Qiwen Zhong, Jie Tian

**Affiliations:** 1Qinghai Key Laboratory of Vegetable Genetics and Physiology, Academy of Agriculture and Forestry Sciences of Qinghai University, Xining 810016, China; 2School of Life Sciences, Lanzhou University, Lanzhou 730000, China

**Keywords:** garlic, reference gene, normalization, RT-qPCR, gene expression

## Abstract

The principal objective of this study was to screen and verify reference genes appropriate for gene expression evaluation during plant growth and development under distinct growth conditions. Nine candidate reference genes were screened based on garlic transcriptome sequence data. RT-qPCR was used to detect the expression levels of the aforementioned reference genes in specific tissues under drought and cold stress. Then, geNorm, NormFinder, BestKeeper, and ReFinder were used to consider the consistency of the expression levels of candidate reference genes. Finally, the stress-responsive gene expression of ascorbate peroxidase (*APX*) was quantitatively evaluated to confirm the chosen reference genes. Our results indicated that there were variations in the abundance and stability of nine reference gene transcripts underneath cold and drought stress, among which *ACT* and *UBC-E2* had the highest transcript abundance, and *18S rRNA* and *HIS3* had the lowest transcript abundance. *UBC* and *UBC-E2* were the most stably expressed genes throughout all samples; *UBC* and *UBC-E2* were the most stably expressed genes during cold stress, and *ACT* and *UBC* were the most stably expressed genes under drought stress. The most stably expressed genes in roots, pseudostems, leaves, and cloves were *EF1*, *ACT*, *HIS3*, *UBC,* and *UBC-E2*, respectively, while *GAPDH* was the most unstable gene during drought and cold stress conditions and in exclusive tissues. Taking the steady reference genes *UBC-E2*, *UBC,* and *ACT* as references during drought and cold stress, the reliability of the expression levels was further demonstrated by detecting the expression of *AsAPX*. Our work thereby offers a theoretical reference for the evaluation of gene expression in garlic in various tissues and under stress conditions.

## 1. Introduction

Garlic (*Allium sativum* L.), a medicinal and edible vegetable, is widely cultivated throughout the world. The shallow root system in garlic seedlings causes its weak abilities which are relevant to drought and cold tolerance [1]. Especially in Qinghai plateau, the climate in spring is characterized by dry weather, low rainfall, and continuous low temperature, which results in decreased yield and low quality of garlic. Therefore, drought and cold stress have become the key problems that are hindering plateau garlic production [2] as plants are often restricted from realizing their full genetic potential [3]. Under cold stress, plants often show dwarfed seedlings and yellow leaves [4]; the damage to the plasma membrane is caused by low solute concentration [5,6]. Drought stress may cause the decomposition of photosynthetic pigments and reactive oxygen species accumulation [7,8], which can damage macromolecules and encourage membrane lipid peroxidation [9]. Plants could resist drought and cold stress by enhancing antioxidant systems [10,11,12], including the trigger of differential expression of antioxidant-related gene ascorbate peroxidase (*APX*) [13,14], transcription of stress-related proteins [15,16,17], and regulation of hormone levels in response to stress [18].

Real time quantitative PCR (RT-qPCR) couples a fluorescent chemical reporter to DNA amplification, thereby monitoring the amount of total product after each PCR cycle and quantitatively analyzing the abundance of transcripts of interest [19]. According to its advantages of outstanding repeatability, excellent sensitivity, and quantitative accuracy [20], it has become an effective method for detecting and quantifying gene expression levels in molecular biology research [21]. Since the quantitative process is easily restricted by the quality of RNA and cDNA, the specificity of primers, and the efficiency of PCR amplification [22], the quantitative analysis of the target gene needs to introduce a stable reference gene for calibration [23,24]. Reference genes used to explore the relative levels of gene expression generally consist of cytoskeletal proteins and essential components involved in the basic biochemical metabolic pathways of cells [25], such as *actin* (*ACT*) [26], *18S somatic ribose RNA* (*18S rRNA*), and *glyceraldehyde-3-phosphate dehydrogenase* (*GAPDH*) [27]. Although the above genes might be stably expressed under any conditions in theory [28], their expression level is not always consistent with the changes in experimental conditions, plant tissues, and physiological states [29]. Therefore, the selection of appropriate reference genes for the calibration and standardization of expression levels according to specific species and different processing methods is necessary to improve the accuracy of RT-qPCR [30]. At present, geNorm, NormFinder, and BestKeeper are common methods for the screening of stable reference genes and authoritative methods in this part [31,32], which were widely used in the stable reference genes of various plants, such as larch [33] and jujube [34]. Despite the screening methods of reference genes being relatively mature and developed, few studies detail the screening of reference genes in plateau garlic and its specific tissues under different treatments, which greatly limits the efficacy of reference genes in garlic.

Toward the evaluation of garlic reference genes, garlic varieties (*A. sativm L. cv.* Ershuizao et al.) were used as materials to screen for stable reference genes at different developmental stages and under salt stress [35,36]. However, knowledge of garlic reference genes during drought and cold stresses remains deficient. In this study, nine widely used reference genes were initially selected from the garlic transcriptomic data of our previous laboratory. The expression levels of reference genes in roots, pseudostems, leaves, and cloves under cold and drought stresses were detected by RT-qPCR. The stability and reliability of the selected reference genes were analyzed and validated by evaluation software. Based on the above, our study will provide a foundation for expanding the selection of reference genes in garlic and analyzing the mechanisms of garlic responses to cold and drought stress.

## 2. Results

### 2.1. RNA Quality Detection and Primer Specificity Analysis of Reference Genes

All the RNA samples were detected by electrophoresis, and the results (Figure 1A) showed that the 5S, 18S, and 28S rRNA bands were intact without obvious tailing, indicating that the extracted RNA had suitable integrity. RNA concentration and quality were determined, and the A260/A280 and A260/A230 of RNA were between 1.9~2.1 and 2.1~2.3, respectively, which could be used for cDNA synthesis.

PCR amplification products detected by agarose electrophoresis were performed on the nine candidate reference genes using the cDNA of different tissues at 0 h treatment as the template. The results (Figure 1B) showed that the fragments of each reference gene were between 100 and 200 bp, which was consistent with the expected fragment sizes. Each product was well amplified with no primer dimers and secondary bands evident, indicating that the selected reference gene primers could be used in subsequent experiments.

### 2.2. Assessment of Primer Amplification Efficiency and Specificity

Based on the garlic transcriptome data, primer sequences for the expression analysis of nine candidate reference genes (*ACT*, *18S rRNA*, *HIS3*, *GAPDH*, *RPS5*, *UBC-E2*, *UBC*, *UBQ*, and *EF1*) were designed (Table 1). The primers’ generated amplicons of the reference gene were between 97–147 bp. The amplification efficiency (E) was generated via the standard curve, accompanied by a linear relationship (R^2^ > 0.93); the amplification efficiency was between 83.24% and 157.45% (Table 1).

### 2.3. RT-qPCR Analysis of Candidate Reference Genes in Garlic

#### 2.3.1. Melting Curve Analysis

cDNA was used as the template for RT-qPCR amplification and the melting curve of the candidate reference gene was further analyzed. The melting curves of the nine candidate reference genes were all single peaks, indicating that the primers were specific and that the amplification curves were reproducible between samples (Figure 2).

#### 2.3.2. Expression Profile Analysis of Candidate Reference Genes

The expression levels of candidate reference genes can be reflected by measuring the cycle threshold (Ct) values based on RT- qPCR. The Ct value is inversely proportional to the gene expression abundance, which indicated that lower Ct value corresponds to higher gene expression abundance. Among all the tested samples under different tissues and stress conditions, the Ct values of nine candidate reference genes were between 24.26 and 34.69. The Ct values of *ACT* and *UBC-E2* were relatively low, while the Ct values of *HIS3* and *18S rRNA* were relatively high (Figure 3).

### 2.4. Expression Stability Analysis of the Reference Genes

#### 2.4.1. geNorm Analysis

The geNorm software determines the expression stability of reference genes by calculating the M value (Expression Stability) of nine candidate reference genes under different experimental conditions and in specific tissues. The program uses M = 1.5 as a threshold; the smaller M value indicated higher stability of the candidate reference gene. In Figure 4, the changes in M value in different conditions and tissues were variant. The expression stability values of all reference samples were evaluated and seven candidate reference genes had M values less than 1.5 (*UBC-E2 < UBC < RPS5 < EF1 < UBQ < 18S rRNA < HIS3*). There were six reference genes with M values lower than 1.5 (*UBC-E2 < UBC < 18S rRNA < HIS3 < EF1 < RPS5*) and nine genes (*UBC-E2 < UBC < ACT < UBQ < 18S rRNA < RPS5 < HI3 < EF1*) that met this criterion under cold and drought stress, respectively. The results of stable reference genes in different tissues showed that *EF1* and *UBC* were the most stable in roots; *ACT* and *UBC* were the most stable in pseudostems; *EF1* and *UBC* were the most stable in leaves; *UBC-E2* and *UBC* were the most stable in cloves. The stability of *GAPDH* was generally the worst in different experimental conditions and tissues.

geNorm determined the optimal number of candidate reference genes by calculating the V_n/n+1_ = 0.15 value to obtain more accurate and reliable results. When V_n/n+1_ is lower than 0.15, there are no candidate reference genes that meet the requirements for correcting the expression level of the target gene. The V_n/n+1_ values of the total samples, samples under drought and cold treatments, roots, leaves, and cloves were all higher than 0.15, and the optimal number of reference genes could not be determined. Therefore, the minimum value of V_n/n+1_ should be selected as the optimal inner parameter under the above conditions. It can be seen from Figure 5 that V_2/3_ = 0.043 < 0.15 in the pseudostem, which indicates that selecting two reference genes in the pseudostem could provide an accurate reference for the detection of the target gene.

#### 2.4.2. NormFinder Analysis

NormFinder determines the stability of the reference gene by the value of gene expression stability (S); the lower S value represented a more suitable reference gene. From the above ranking results, the most stable reference genes in all samples were *UBC-E2* and *EF1* and the most stable reference genes under cold and drought were *HIS3*, *UBC-E2, EF1*, and *ACT*. Among different tissues, the most stable reference genes in roots, pseudostems, and cloves were *ACT*, *18S rRNA*, *UBQ*, *HIS*3, *UBC*, *EF1,* and *UBC-E2* (Table 2).

#### 2.4.3. BestKeeper Analysis

BestKeeper evaluates the stability of candidate reference genes by the standard deviation (SD) and coefficient of variation (CV). Lower SD and CV indicate better stability of reference genes. As shown in Table 3, the stabilities of *HIS3* and *18S rRNA* were the best among all samples; the stabilities of *UBC-E2* and *HIS3* were the highest under cold stress; the expression of both *HIS3* and *RPS5* were relatively stable under drought stress. The stability in different tissues of garlic was evaluated and the transcript stability in the root was ranked as *ACT* and *HIS3*; the stability of pseudostem was best in *18S rRNA* and *HIS3*; the stability of leaf was best in *18S rRNA* and *GAPDH*, and the stability in cloves was best in *UBC-E2* and *18S rRNA*. The stability of gene transcripts varied greatly under different experimental conditions, so the reference gene needs to be carefully considered depending on different experimental conditions.

#### 2.4.4. ReFinder Analysis

Finally, ReFinder was used to comprehensively rank reference genes in different treatment conditions and tissues. *UBC* and *UBC-E2* were the most stably expressed genes in all samples, while *UBC* and *UBC-E2* were the most stably expressed genes under cold stress, and *ACT* and *UBC* were the most stably-expressed genes under drought stress. In different tissues, the most stably expressed genes in roots, pseudostems, leaves, and cloves were *EF1* and *ACT*, *HIS3* and *UBC*, *UBC* and *EF1*, and *UBC-E2* and *UBC*, respectively. *GAPDH* was the most unstable gene expressed in drought, cold stress, and different tissues (Table 4).

### 2.5. Stability Verification of CANDIDATE Reference Genes

To verify the reliability of candidate reference genes, the gene expression of *APX* was calculated using the identified reference genes. APX, one of the important antioxidant enzymes in plant reactive oxygen species’ metabolism, especially the key enzyme for scavenging H_2_O_2_ in the chloroplast, has been shown to be related to stress tolerance. The relatively stable reference genes (*UBC-E2* and *UBC*, and *ACT* and *UBC*, respectively) and unstable reference genes (*GAPDH*) were used to detect the expression of *AsAPX* in garlic at 12 h under cold and drought stress, thereby verifying the stability and accuracy of the selected reference genes (Figure 6). The expression levels of *AsAPX* under drought and cold stress were the same when the stable reference genes were used, but the expression levels determined by the unstable reference gene *GAPDH* were significantly different. Thus, these data further prove the reliability of the experimental results of this study.

## 3. Discussion

With the advancement of molecular biology, research on the expression of functional genes in plants has become a hot topic of research [37]. Although gene expression can be detected by genome and transcriptome sequencing, RT-qPCR is still a vital method due to its advantages of high sensitivity, flexible detection, and excellent efficiency [38,39]. Nevertheless, the accuracy of RT-qPCR detection results is often affected by the specificity of primers, length of amplified products, quality of RNA, and consistent reference genes [40,41]. Meanwhile, the use of common reference genes such as *18S rRNA* and *ACTIN* without screening also reduce the accuracy of quantitative results and even lead to incorrect conclusions [42].

To explore the changes in gene expression levels in organisms, the screening of stable reference genes is essential. geNorm, NormFinder, and BestKeeper were widely used in the evaluation of the transcript stability of reference genes but determining the stability of most reference genes could be dependent on different software parameters [43]. In this study, the most stable reference genes screened by geNorm under cold stress were *UBC-E2* and *UBC*, while *HIS3*, *UBC-E2*, and *HIS3* were the reliable reference genes as determined by NormFinder and BestKeeper, respectively. geNorm, NormFinder, and BestKeeper found that the most stable reference genes were *UBC-E2*, *UBC*, *EF1*, *ACT*, *HIS3,* and *RPS5* under drought stress. Similar results were obtained in Cynomorium [44] and Chrysanthemum [45]. To improve the detection accuracy of the target gene and resolve the diversities between different software, Vandesompele [46] recommends using the V value (Pairwise variation value) generated from geNorm as the threshold to further reduce errors by using a combination of multiple reference genes. This program usually takes V_n/n+1_ equal to 0.15 as a threshold. When the threshold was less than 0.15, there were no candidate reference genes to satisfy the requirements of correcting the expression of the target gene. Due to 0.15 being an ideal value, which mainly depends on the number of genes and the type of test samples, there were still large differences in the accuracy of screening results [47]. When V = 0.15 was used for calibration in garlic, only the value of V_2/3_ in cloves was lower than 0.15 and two genes were needed to achieve the purpose of accurate calibration. However, more than two reference genes were required to be used together in different experimental treatments and tissues (Figure 2), which not only amplifies the experimental error but also increases the experimental budget [36]. Therefore, in order to avoid errors caused by differences in software analysis and human selection, integrated analysis by comprehensively evaluating the results obtained from different software was used as an important means to reflect the stability of each reference gene and reduce the differences caused by software evaluation. ReFinder software was used to comprehensively evaluate the results of the three conventional softwares based on previous studies, and the most stably expressed genes under cold and drought stress were *UBC* and *UBC-E2*, and *ACT* and *UBC*, respectively (Table 4). Some researchers have also used ReFinder technology to evaluate the stability of reference genes in different varieties of pears [48] and rice [49], which means that the reference genes comprehensively screened by ReFinder software were stable and reliable.

In garlic, Liu [35] screened *CYP* as the most suitable reference gene for abiotic stress conditions but lacked specific reference genes for cold stress. Wang [36] identified *ACTIN* and *SAND* as stable reference genes in garlic under salt stress. However, there are still relatively few stable reference genes in garlic under plateau climate stress, which greatly limits genetic resources in garlic for the detection of gene expression levels. In our study, the stable reference genes in garlic under cold and drought stress were U*BC-E2*, *UBC* and *ACT*, *and UBC*, respectively. Similarly, the *UBC* and *ACTIN* genes in *Psathyrostachys huashanica* under abiotic and biotic stresses were confirmed to be consistently expressed in all samples [50]. Schmidt combined geNorm, NormFinder, BestKeeper, and other software to screen out eight candidate reference genes of tobacco and found that *UBC-E2* was a highly stable reference gene in different tissues and throughout the entrained circadian rhythm(s) [51]. Wang screened for the most stable reference gene in *Iris germanica* L., which was *UBC* [52]. *Actin* is a reference gene that has been widely used in many plant species at present [53], which could be proved in Kosteletzkya Virginica’s research [54]. However, *actin* was demonstrated to be the best-performing reference gene in all test samples under drought stress in this study. Glyceraldehyde-3-phosphate dehydrogenase (*GAPDH*), as a key component in plant response to abiotic stress, has unique advantages as a probe due to its low homology across phyla [55]. *GAPDH* was used as a stable reference gene in cotton [56] and Arabidopsis [57], providing an important reference for the expression of key genes. However, in this study, *GAPDH* had the lowest expression stability in different treatments and tissues, which was consistent with the screening results of reference genes in garlic under salt stress by Wang [36], indicating that *GAPDH* was not suitable as a reference gene for garlic.

*APX* is one of the key antioxidant-related genes, which plays the curial role during reactive oxygen species (ROS) signal transduction in plants [58,59,60]. Both Verma [61] and Lee [62] have demonstrated that *APX* was directly involved in ROS detoxification, protecting cells from oxidative bursts. ROS signal transduction is not steady under abiotic stress, but the reference gene can be stably expressed, which guarantees accurate detection of ROS inducing *APX* differential expression level [63]. For garlic, Wani [64] found that the transcript expression of cold responsive genes (*APX*, *GR*) was detected to be upregulated under cold stress. Fones [65] evaluated the suitability of the selected reference gene by analyzing the expression levels in *APX*. Compared with the method of Wang [36] to verify the selected reference gene with the garlic *AO* gene, the stable reference genes in this study were verified by *AsAPX* and were found to be excellent reference genes in garlic, which is consistent with the identification of reference genes in kiwifruit through *APX* [66].

## 4. Conclusions

The identification of reliable reference genes is the prerequisite for qualifying gene expression under specific experimental conditions. In the present study, a systematic analysis validates a set of reference genes by RT-qPCR in garlic various tissues subject to stressful environments. Based on the above results, *ACT*, *UBC,* and *UBC-E2* were recommended as the most stable reference genes in garlic under drought and cold stress. Current research may help to accurately quantify the target genes in garlic and reveal the molecular mechanisms of garlic stress tolerance.

## 5. Materials and Methods

### 5.1. Plant Material

*Allium sativum* L. variety ‘Ledu Purple Skin Garlic’, as a special Qinghai-plateau vegetable, preserved at the Horticulture Institute of Academy of Agriculture and Forestry Sciences in Qinghai University, Xining, China, was used as experimental material in the present study. Garlic cloves released from dormancy with similar size were selected and planted into plastic pots (top diameter 10cm, bottom diameter 8 cm,) filled with commercial substrate (peat: perlite = 1: 1). Then, the plants were cultured in a growth chamber with a 14-h artificial light (25 °C): 10-h dark (10 °C) cycle and 65–75% relative humidity. Cultivated until the garlic seedling stage (the seedling height reached approximately 10 cm), the treatments were performed as follows: for drought stress, seedlings were cultivated by controlling substrate moisture content (45–55%); for cold stress, seedlings were cultivated at the growth chamber with temperature of 4 °C. Different parts of garlic (root, pseudostem, leaf, clove) were then collected at 0 h, 1 h, 4 h, 12 h, and 1 d after the respective treatments. Each experiment was completed with three replicates, each comprised of five seedlings per treatment. All collected samples were immediately frozen in liquid nitrogen and stored at −80 °C until analyzed.

### 5.2. Total RNA Isolation and cDNA Synthesis

Total RNA was extracted from frozen samples using the Trizol Reagent (DP405, Tiangen Biotech, Beijing, China). The integrity and quality of extracted RNA were evaluated by 1% agarose gel electrophoresis and a TGem micro-spectrophotometer (OSE-260, Tiangen Biotech, Beijing, China). Only the A260/A280 value of the total RNA of each sample ranging between 1.8–2.1 meet the requirements of subsequent experiments. The above RNA samples as a template referring to the cDNA synthesized were carried out using the Fast Quant cDNA First Strand Synthesis Kit (KR106, Tiangen Biotech Beijing, Beijing, China). The products were stored at −20 °C until analyzed.

### 5.3. Primer Design

The nine candidate genes that were selected for screening based on their role as reference genes in other plants were aligned with the garlic transcriptome established by our group, including *ACT*(*Actin*), *EF1*(*elongationfactor1*), *UBC-E2*(*Ubiquitin-conjugating enzyme-E2*), *GAPDH*(*Glyceraldehyde-3-phosphatedehydrogenase*), *HIS3* (*Histone H3*), *RPS5*(*ribosomalprotein S5*), *UBC*(*Ubiquitin-conjugating enzyme*), *UBQ*(*Polyubiquitin*), *18S rRNA*(*18S ribosomal RNA*). All RT-qPCR primers were designed with the Primer 5.0.

### 5.4. PCR and RT-qPCR Analysis of Reference Genes

According to the instructions provided by Tiangen Biotech, Beijing, China, for the Super Real Fluorescence Quantitative Premix Reagent-Enhanced Kit (SYBR Green, FP205), we set the mixture into the Bio-RadiQ5 Real-Time Fluorescence Quantitative Instrument for PCR amplification. The reaction mixture was comprised of 10 μL 2×SYBR Green RT-qPCR Master Mix, 8.2 μL ddH_2_O, 1 μL cDNA, and 0.4 µL each of forward and reverse amplification primers for a final volume of 20 μL. The amplification conditions were as follows: Pre-denaturation at 95 °C for 5 min, denaturation at 95 °C for 10 s, TM (53–63 °C) annealing, extension for 30 s, 40 cycles. A blank control lacking a template was also included at the same time to ensure amplification quality and gene expression levels for each sample were determined based on three replicates. The amplification specificity of each pair of primers was evaluated by 1% agarose gel electrophoresis and melting curve analysis, and the PCR amplification efficiency (E) and correlation coefficient (R^2^) were calculated using a 5-fold cDNA dilution series.

### 5.5. Analysis of Reference Gene Expression Stability

The cycle threshold (Ct) value was recorded for each RT-qPCR experiment. Through geNorm [46], NormFinder [67], and BestKeeper [68] analyses, we analyzed the transcript stability of garlic candidate reference genes under drought and cold stress. Among them, GeNorm and NormFinder used the 2^-∆∆CT^ method [37] to convert the CT value into the relative expression value for their respective analysis, while BestKeeper used the original CT value to evaluate consistency. Finally, ReFinder [69] was used for comprehensive evaluation [38] and the most stable reference gene(s) were determined. According to the selected reference genes, the expression patterns of *APX* during garlic development under drought and cold stress were analyzed to further verify the selected stable reference genes.

## Figures and Tables

**Figure 1 plants-12-00763-f001:**
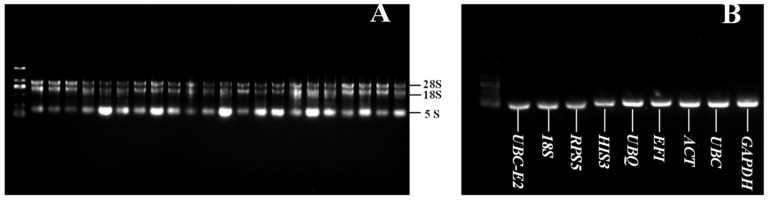
Garlic RNA samples (**A**) and amplification products of candidate reference genes (**B**) detected by agarose electrophoresis.

**Figure 2 plants-12-00763-f002:**
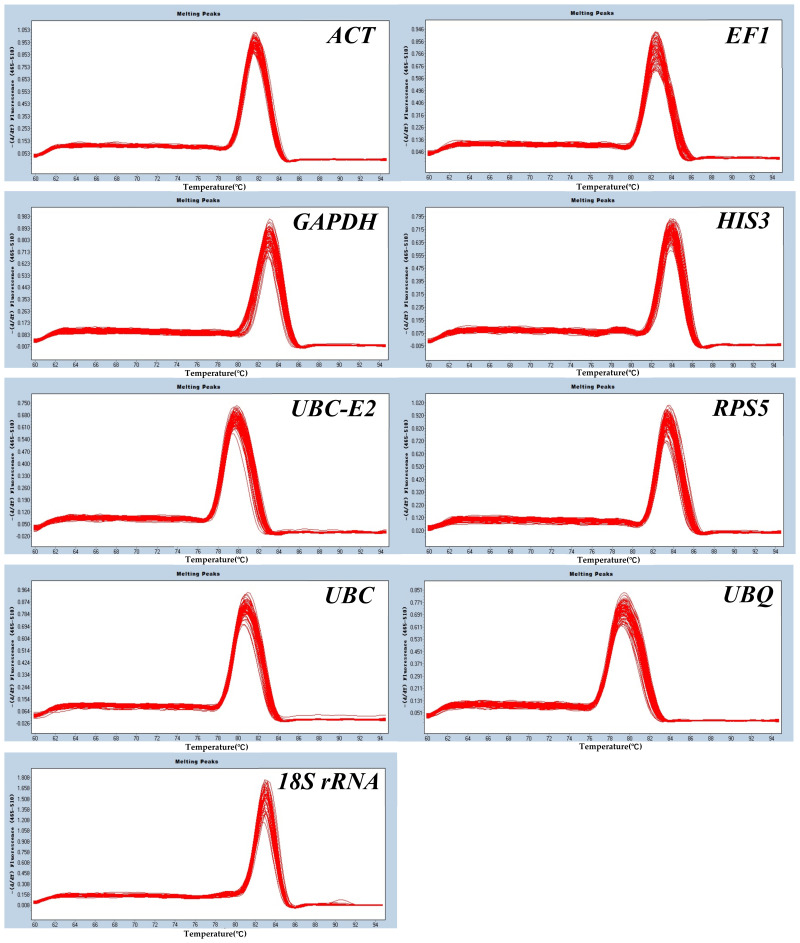
RT−qPCR melting curves of the nine candidate reference genes in garlic.

**Figure 3 plants-12-00763-f003:**
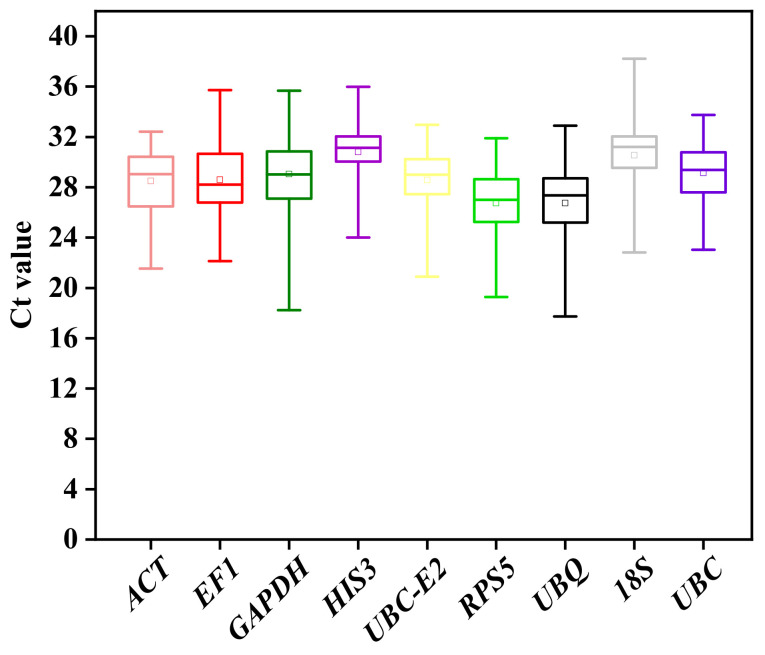
Distribution of Ct values of the nine candidate reference genes in garlic among all the tested samples. The boxes indicate the 25th and 75th percentiles. The line across the box and the inner square in each box indicate the median and mean Ct values, respectively.

**Figure 4 plants-12-00763-f004:**
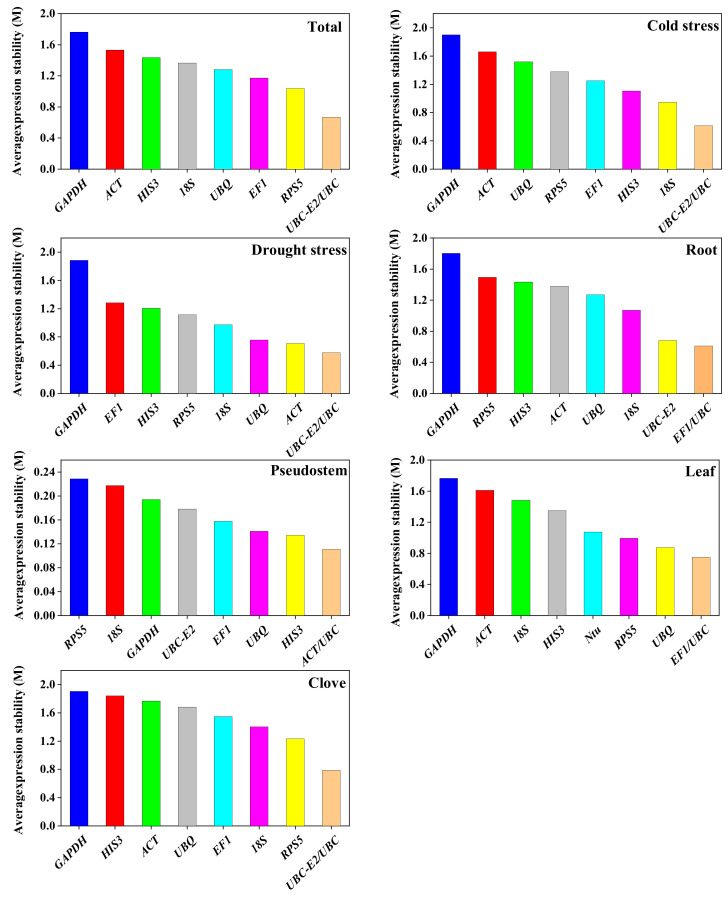
Mean expression stability values (M) of candidate reference genes. Note: The most stable genes are listed on the right and the most unstable genes are listed on the left.

**Figure 5 plants-12-00763-f005:**
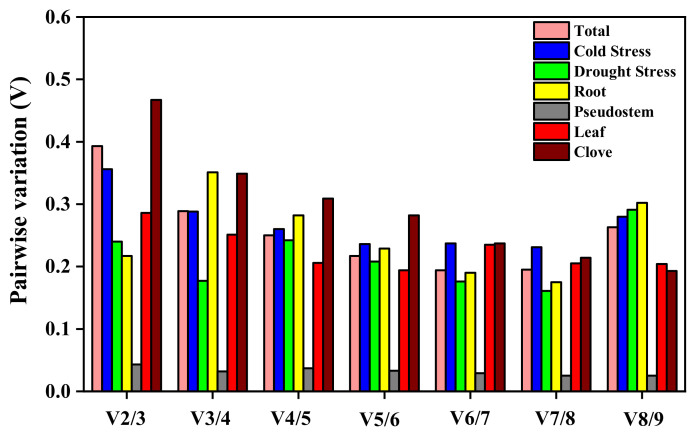
Determination of optimal parameters for candidate reference genes.

**Figure 6 plants-12-00763-f006:**
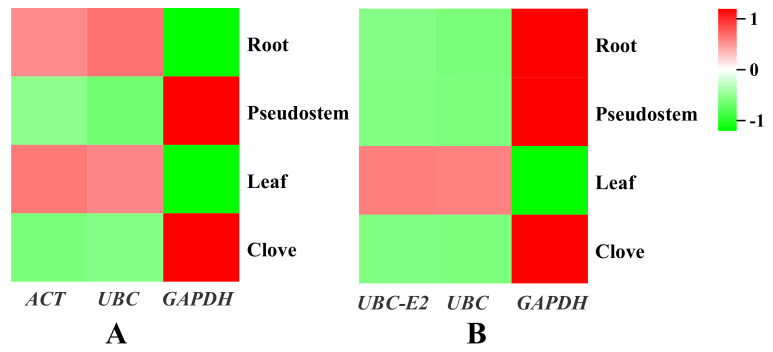
*AsAPX* expression in different parts of garlic under drought(**A**) and cold(**B**)stress.

**Table 1 plants-12-00763-t001:** Information on the candidate reference genes of garlic.

Genename	Primer Sequence (Forward/Reverse)	Amplicon Length (bp)	Tm (℃)	E (%)	Slope	R^2^
*ACT*	ATTAGTGTCGCCATTCTT	117	54	91.25413	−3.551	0.999
*Actin*	TTGACGCACATTACCATC					
*18S rRNA(18 S)*	CGCTGGTGGCGTAGTTGT	101	58	109.1765	−3.12	0.996
*18S ribosomal RNA*	TGGGAAGGGTGGTTTGTG					
*HIS3*	CCCGTCACAGAGGAAAGA	121	63	157.4409	−2.435	0.936
*HistoneH3*	GAGCAGCAGGGATAAGCA					
*GAPDH* *Glyceraldehyde-3-phosphatedehydrogenase*	CCCTGGCAAAGGTGAT	106	53	96.06391	−3.42	0.999
	AAGGCAGTTGGTGGTG					
*RPS5*	TACCGACCAGAACCCTA	103	53	90.62968	−3.569	0.998
*RibosomalproteinS5*	CTGCCTGACGCCTAAC					
*UBC-E2*	CGGTTTGTATGAATGTGC	97	54	97.83189	−3.375	0.992
*Ubiquitin-conjugating enzyme E2*	TTAGGGTAAGAAAGGAGTTG					
*UBC*	TTCGGGTTCGGTTTGTAT	105	56	83.23965	−3.802	0.996
*Ubiquitin-conjugating enzyme*	TTAGGGTAAGAAAGGAGTTGAG					
*UBQ*	GGAAGATGGCAGAACG	139	50	94.88171	−3.451	0.999
*Polyubiquitin*	GCACAAGATGAAGGGTA					
*EF1*	GCATAAAGAAGGAGGGT	147	56	85.79506	−3.717	0.992
*elongationfactor1*	CTGGTTCGGCAGTAAG					

**Table 2 plants-12-00763-t002:** Stable values of nine candidate reference genes in different treatments and tissues by NormFinder.

Rank	Total		Cold Stress		Drought Stress		Root		Pseudostem		Leaf		Clove	
	Gene	Stability	Gene	Stability	Gene	Stability	Gene	Stability	Gene	Stability	Gene	Stability	Gene	Stability
**1**	*UBC-E2*	0.035	*HIS3*	0.48	*EF1*	0.566	*ACT*	0.582	*UBQ*	0.063	*UBC*	0.482	*UBC-E2*	0.7
**2**	*EF1*	0.041	*UBC-E2*	0.633	*ACT*	0.579	*18S rRNA*	0.645	*HIS3*	0.072	*EF1*	0.527	*UBC*	0.722
**3**	*HIS3*	0.042	*UBC*	0.692	*UBC*	0.607	*EF1*	0.72	*ACT*	0.072	*UBC-E2*	0.564	*RPS5*	0.73
**4**	*UBQ*	0.046	*EF1*	0.832	*UBC-E2*	0.65	*HIS3*	0.744	*UBC-E2*	0.073	*UBQ*	0.637	*EF1*	0.772
**5**	*18S rRNA*	0.047	*UBQ*	0.901	*RPS5*	0.657	*UBC-E2*	0.802	*UBC*	0.083	*RPS5*	0.657	*UBQ*	0.837
**6**	*ACT*	0.05	*ACT*	0.923	*UBQ*	0.658	*UBC*	0.81	*GAPDH*	0.085	*18S rRNA*	0.784	*18S rRNA*	0.857
**7**	*RPS5*	0.051	*18S rRNA*	1.03	*HIS3*	0.689	*UBQ*	0.923	*EF1*	0.096	*HIS3*	0.882	*HIS3*	0.985
**8**	*UBC*	0.062	*RPS5*	1.132	*18S rRNA*	0.772	*RPS5*	1.052	*18S rRNA*	0.108	*GAPDH*	0.943	*GAPDH*	1.006
**9**	*GAPDH*	0.069	*GAPDH*	1.341	*GAPDH*	1.525	*GAPDH*	1.589	*RPS5*	0.113	*ACT*	1.09	*ACT*	1.009

**Table 3 plants-12-00763-t003:** Stable values of nine candidate reference genes in different treatments and tissues in BestKeeper.

Rank	Total			Cold Stress			Drought Stress			Root			Pseudostem			Leaf			Clove		
	Gene	SD	CV	Gene	SD	CV	Gene	SD	CV	Gene	SD	CV	Gene	SD	CV	Gene	SD	CV	Gene	SD	CV
**1**	*HIS3*	0.773	2.451	UBC-E2	0.787	2.751	*HIS3*	0.711	2.271	*ACT*	0.309	1.07	*18S rRNA*	0.367	1.142	*18S rRNA*	0.344	1.073	*UBC-E2*	0.742	2.45
**2**	*18S rRNA*	0.845	2.676	*HIS3*	0.927	2.921	*RPS5*	0.752	2.824	*HIS3*	0.744	2.375	*HIS3*	0.554	1.762	*GAPDH*	0.617	2.273	*18S rRNA*	0.784	2.478
**3**	*UBC-E2*	1.112	3.775	*UBC*	0.933	3.23	*18S rRNA*	0.801	2.564	*18S rRNA*	0.83	2.721	*UBC-E2*	0.621	2.137	*HIS3*	0.625	1.982	*UBC*	0.846	2.761
**4**	*UBC*	1.26	4.218	*UBQ*	0.945	3.524	*EF1*	0.858	3.047	*UBQ*	1.373	5.065	*GAPDH*	0.659	2.39	*RPS5*	0.956	3.389	*UBQ*	1.17	4.149
**5**	*RPS5*	1.284	4.648	*18S rRNA*	0.97	3.081	*ACT*	0.885	2.944	*GAPDH*	1.525	5.046	*UBC*	0.961	3.218	*UBC-E2*	1.079	3.623	*HIS3*	1.267	3.98
**6**	*UBQ*	1.321	4.747	*RPS5*	1.329	4.887	*UBC-E2*	0.994	3.419	*RPS5*	1.531	5.717	*RPS5*	1.274	4.647	*UBC*	1.12	3.726	*RPS5*	1.311	4.672
**7**	*ACT*	1.484	5.066	*EF1*	1.513	5.357	*UBC*	1.023	3.494	*EF1*	1.588	5.592	*UBQ*	1.295	4.564	*EF1*	1.287	4.385	*GAPDH*	1.391	4.584
**8**	*GAPDH*	1.67	5.794	*ACT*	1.677	6.149	*UBQ*	1.145	4.227	*UBC-E2*	1.609	5.6	*ACT*	2.037	6.956	*UBQ*	1.413	5.113	*ACT*	1.397	4.719
**9**	*EF1*	1.712	5.84	*GAPDH*	1.883	6.604	*GAPDH*	2.046	7.101	*UBC*	1.723	5.968	*EF1*	2.042	7.092	*ACT*	1.951	6.66	*EF1*	1.596	5.195

**Table 4 plants-12-00763-t004:** Stability values of nine candidate reference genes in different treatments and tissues in BestKeeper.

Rank	Total		Cold Stress		Drought Stress		Root		Pseudostem		Leaf		Clove	
	Genes	Rank	Genes	Rank	Genes	Rank	Genes	Rank	Genes	Rank	Genes	Rank	Genes	Rank
1	*UBC*	1.41	*UBC*	1.68	*ACT*	1.57	*EF1*	1.97	*HIS3*	1.41	*UBC*	1.50	*UBC-E2*	1.00
2	*UBC-E2*	1.86	*UBC-E2*	1.73	*UBC*	2.30	*ACT*	2.34	*UBC*	1.86	*EF1*	2.30	*UBC*	1.68
3	*RPS5*	3.94	*HIS3*	1.86	*UBC-E2*	2.71	*18S rRNA*	3.13	*UBC-E2*	2.06	*UBQ*	3.83	*RPS5*	3.71
4	*HIS3*	3.96	*18S rRNA*	4.36	*HIS3*	4.30	*UBC*	3.20	*GAPDH*	4.23	*RPS5*	4.00	*18S rRNA*	4.16
5	*UBQ*	4.05	*EF1*	4.95	*18S rRNA*	4.61	*UBC-E2*	3.46	*18S rRNA*	4.82	*HIS3*	4.30.	*EF1*	5.03
6	*18S rRNA*	4.36	*UBQ*	5.38	*UBQ*	4.76	*HIS3*	5.12	*UBQ*	5.48	*GAPDH*	4.56	*UBQ*	5.96
7	*EF1*	6.05	*RPS5*	6.96	*RPS5*	5.18	*UBQ*	5.18	*ACT*	7.24	*UBC-E2*	5.23	*HIS3*	6.05
8	*ACT*	7.74	*ACT*	7.74	*EF1*	7.11	*RPS5*	8.00	*EF1*	8.24	*18S rRNA*	6.26	*ACT*	7.20
9	*GAPDH*	9.00	*GAPDH*	9.00	*GAPDH*	9.00	*GAPDH*	9.00	*RPS5*	8.45	*ACT*	9.00	*GAPDH*	9.00

## Data Availability

Not applicable.

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
