# Peer review of "Screening and Verification of Reference Genes for Analysis of Gene Expression in Garlic (Allium sativum L.) under Cold and Drought Stress"

_plants, 2023, doi:10.3390/plants12040763_

Round 1

Reviewer 1 Report

In order to verify reference genes in garlic responses to cold and drought stress, nine candidate reference genes were screened based on garlic transcriptome sequence data. Then, the expression levels of reference genes in roots, pseudostems, leaves, and cloves under cold and drought stress were detected by RT-qPCR. The selected reference genes were validated and analyzed. The results indicated that there were variations in the abundance and stability of nine reference gene transcripts under cold and drought stress conditions. UBC-E2, UBC and ACT, UBC were recommended stable reference genes under the drought and cold stress conditions. However, the MS is suggested to revise to satisfy the journal requirement for publication.

1.       ROS generation always occurs during plant responses to environmental stresses. However, it is suggested to supplement additional discussions with the signal transduction and reference genes responses.

2.       The abscissa of each gene in Figure 4 is not a continuous variable, so it is recommended to use a histogram instead of a continuous curve for the values of dependent variables in this figure.

3.       Extensive editing of English language and style required.

Reviewer 2 Report

Dear Editor and Authors

There are some comments and recommendations for the reviewed manuscript. All findings are listed below:

1.     The numbers of amplification efficiency mentioned in the text (line 108) are not in accordance with the values given in Table 1.  

2.     Chapter 2.3.2, Line 124 - it would be desirable to explain the relation between Ct values and gene expression.

3.     Chapter 2.5 - In this chapter, the authors describe the suitability of selected reference genes for evaluating ascorbate peroxidase gene expression. The result of gene expression should be shown in Figure 6, which is missing in the manuscript. Therefore it is difficult to assess the reliability of selected reference genes for evaluation of ascorbate peroxidase gene expression under different stress conditions.

4.     I recommend adding a separate chapter "Conclusion" in the manuscript, which should be part of the manuscript according to the manuscript preparation instructions.

Round 2

Reviewer 1 Report

The revised manuscript has been sufficiently improved to warrant publication in Plants. Besides, it is suggested to change the Fig. 4 with color picture to ensure the consistency of the full text style.

Reviewer 2 Report

Dear Editor and Authors,

all my former findings were sufficiently considered in the text. I recommend publishing the manuscript.